# Active Transport of Hepatotoxic Pyrrolizidine Alkaloids in HepaRG Cells

**DOI:** 10.3390/ijms22083821

**Published:** 2021-04-07

**Authors:** Anne-Margarethe Enge, Florian Kaltner, Christoph Gottschalk, Albert Braeuning, Stefanie Hessel-Pras

**Affiliations:** 1Department of Food Safety, German Federal Institute for Risk Assessment, Max-Dohrn-Str. 8–10, 10589 Berlin, Germany; anne-margarethe.enge@bfr.bund.de (A.-M.E.); albert.braeuning@bfr.bund.de (A.B.); 2Chair of Food Safety and Analytics, Veterinary Faculty, Ludwig Maximilian University of Munich, Schoenleutnerstr. 8, 85764 Oberschleissheim, Germany; florian.kaltner@ls.vetmed.uni-muenchen.de (F.K.); christoph.gottschalk@ls.vetmed.uni-muenchen.de (C.G.)

**Keywords:** pyrrolizidine alkaloids, structure-dependency, uptake, hepatic transporter

## Abstract

1,2-unsaturated pyrrolizidine alkaloids (PAs) are secondary plant metabolites occurring as food contaminants that can cause severe liver damage upon metabolic activation in hepatocytes. However, it is yet unknown how these contaminants enter the cells. The role of hepatic transporters is only at the beginning of being recognized as a key determinant of PA toxicity. Therefore, this study concentrated on assessing the general mode of action of PA transport in the human hepatoma cell line HepaRG using seven structurally different PAs. Furthermore, several hepatic uptake and efflux transporters were targeted with pharmacological inhibitors to identify their role in the uptake of the PAs retrorsine and senecionine and in the disposition of their *N*-oxides (PANO). For this purpose, PA and PANO content was measured in the supernatant using LC-MS/MS. Also, PA-mediated cytotoxicity was analyzed after transport inhibition. It was found that PAs are taken up into HepaRG cells in a predominantly active and structure-dependent manner. This pattern correlates with other experimental endpoints such as cytotoxicity. Pharmacological inhibition of the influx transporters Na^+^/taurocholate co-transporting polypeptide (SLC10A1) and organic cation transporter 1 (SLC22A1) led to a reduced uptake of retrorsine and senecionine into HepaRG cells, emphasizing the relevance of these transporters for PA toxicokinetics.

## 1. Introduction

Pyrrolizidine alkaloids are secondary plant metabolites that are estimated to be produced by more than 6000 plant species and to be present in about 3% of the world’s flowering plants [1,2]. So far, more than 660 different pyrrolizidine alkaloids and their *N*-oxides have been identified, and it is assumed that roughly half of these are hepatotoxic, making these compounds the most widely occurring natural toxins. Pyrrolizidine alkaloids are found as food contaminants in e.g., tea, honey, and herbs, as well as in feed, turning them into a relevant concern for risk assessment [3,4,5,6,7]. Apart from that, these alkaloids can also be found in dietary supplements and herbal preparations such as traditional medicinal herbs [8,9,10,11,12,13].

Chemically, pyrrolizidine alkaloids share a common basic structure. They are esters of 1-hydroxymethylpyrrolizidine, the necine base, that can be esterified at the hydroxyl groups at ring positions C-7 and/or C-9 with one or two aliphatic mono- or dicarboxylic acids, so-called necic acids. Pyrrolizidine alkaloids can be classified according to their necine base into four different main structure types: retronecine-, heliotridine-, otonecine- and platynecine-type [7]. Furthermore, pyrrolizidine alkaloids are also subdivided by their degree of esterification into monoesters, open-chained diesters and cyclic diesters (chemical structures of representatives depicted in Figure 1). Except for the platynecine-type, all pyrrolizidine alkaloids possess a 1,2-unsaturated necine base (abbreviated in the following as PAs), which is considered to be essential for PA-mediated toxicity [5,7].

It is widely accepted that PAs require metabolic activation in the liver to cause hepatotoxicity [14,15,16]. During hepatic phase I metabolism, PAs are hydroxylated by cytochrome P450 (CYP) enzymes primarily belonging to the 3A and 2B subfamilies [17]. Thus, 3- or 8-hydroxynecine derivatives are formed, which then spontaneously dehydrate to their corresponding electrophilic pyrrolic esters. These reactive esters then hydrolyze to the less reactive and more stable (±)-6,7-dihydro-7-hydroxy-1-hydroxymethyl-5*H*-pyrrolizidine (DHP) [1,17,18]. The generation of both, DHP and pyrrolic esters derivatives, can lead to the formation of protein and DNA adducts, causing acute and chronic toxic effects [5,19]. However, the reactive metabolites can also be detoxified. PAs can hydrolyze into their necine base and necic acids [20]. *N*-oxidation, leading to the formation of the PA *N*-oxides (PANOs), is also considered to be a pathway of detoxification [1]. Additionally, phase II conjugation reactions with glutathione (GSH), 3′-phosphoadenosine-5′-phosphosulfate (PAPS) or glucuronic acid can also occur, resulting in more excretable conjugates [21]. It is well-documented that the structurally diverse PAs show different genotoxic and cytotoxic effects [22].

So far, PA-mediated toxic effects and the role of xenobiotic-metabolizing enzymes in PA activation have been extensively studied. However, until today there are still uncertainties as to why exactly PAs show varying potencies. One aspect that has been barely taken into account is hepatic uptake and transport. These toxicokinetic properties might contribute significantly to structure-dependent differences in the effects of PAs.

Hepatic transporters belong to the two super-families of ATP-binding cassette (ABC) transporters and solute carrier (SLC). SLC transporters mostly act as uptake transporters through facilitated biological diffusion or secondary active transport to move diverse solutes across the sinusoidal membrane [23,24]. Prominent representatives are transporters belonging to the sodium-independent transporters, such as organic anion transporting polypeptides (SLC21/SLCO) and the organic cation transporter 1 (SLC22A1) from the organic anion/cation/zwitterion transporter family (SLC22), and the hepatocyte-specific Na^+^/taurocholate co-transporting polypeptide (SLC10A1) [25,26,27].

In contrast, ABC transporters act as primary active efflux carrier that excrete endogenous compounds and xenobiotics under energy expense into one specific direction independent of the concentration gradient [28]. The majority of hepatic ABC transporters are found in the bile canalicular membrane. However, exceptions are members of the multidrug resistance-associated protein familylike ABCC1 and ABCC3, which efflux their substrates into the blood rather than the bile. Prominent apical ABC transporters include the highly expressed multidrug-resistance protein 1 (ABCB1), ABCC2, the bile salt export pump (ABCB11), and the breast cancer resistance protein (ABCG2) [29,30,31].

Until today, only little is known about specific influx mechanisms of PAs and the efflux of PANOs. Therefore, this study aimed to extend the scientific knowledge on PA toxicity by analyzing liver-specific uptake and transport processes of structurally different PAs. A selective uptake and transport along with metabolism-dependent processes is assumed to confer the structure-dependent different toxic effects of PAs. Therefore, SLC22A1 was analyzed as it was described to be relevant for monocrotaline and retrorsine uptake before [32,33]. SLC10A1 was investigated because of its hepatocyte specificity. Likewise, the co-expressed efflux transporters ABCB1 and ABCG2 were considered to be necessary for PANO distribution, because ABCB1 mediates apical excretion of the PA echimidine in the human small intestinal cell line Caco-2 [34] and ABCG2 shares many overlapping substrates and inhibitors with ABCB1 [35,36]. Also, ABCC transporters were investigated for their role in PANO excretion as they are known to transport a variety of structurally diverse xenobiotics [37,38,39].

## 2. Results

### 2.1. Active and Structure-Dependent Uptake of PAs in HepaRG Cells

A first step in assessing PA transport processes in HepaRG cells was to ascertain the general mode of action of transport. As passive permeability can mask active uptake mechanisms in hepatocytes, incubations at 4 °C can be used as a first qualitative approach to evaluate active transport processes [40]. This temperature method relies on the assumption that energy-driven active transport systems are not functional at 4 °C.

The decrease of PA content and increase of PANO content in the cell culture medium of HepaRG cells incubated either at physiological 37 °C or at 4 °C over the time period of 24 h was measured for echimidine, echimidine *N*-oxide, heliosupine, heliosupine *N*-oxide, intermedine, intermedine *N*-oxide, lycopsamine, lycopsamine *N*-oxide, retrorsine, retrorsine *N*-oxide, senecionine, senecionine *N*-oxide and senkirkine with LC-MS/MS (Figure 2). The selected PAs were chosen as they represent the main structural characteristics of PAs: different necine base types (retronecine-type, heliotridine-type and otonecine-type) and the different necic acids monoester, open-chain diester and cyclic diester (Figure 1).

HepaRG cells incubated at 4 °C showed a substantially higher amount of PA content in their supernatant during all time points compared to cells incubated at 37 °C (Figure 2c). The viability of the cells incubated at non-physiological conditions was monitored and did not fall below 80% after 24 h.

At 37 °C, the reduction of PA contents in the supernatants seemed to be structure-dependent, with the monoesters lycopsamine and intermedine showing only little reduction after 24 h and the diesters heliosupine and senecionine displaying the highest decline of mother compound in the cell culture medium (Figure 2a). In between, senkirkine, retrorsine, and echimidine were ranked in this order. It is noticeable that the grouping of these PAs correlates with recently published cytotoxicity data by Glück et al., (2020) [41]. The amount of formed retrorsine *N*-oxide, echimidine *N*-oxide and senecionine *N*-oxide increased with time (Figure 2b) and most strikingly for the cyclic diesters senecionine and retrorsine, as it was observed before [42]. Senkirkine is a representative of the otonecine-type PAs and did not form PANOs [20]. No PANOs were formed in cold-incubated cells except for senecionine *N*-oxide (<1% of the applied mother compound; Figure 2d).

PAs are known to be very stable [43]. Therefore, a decrease of the mother compound in the cell culture medium is assumed to be equivalent to PA uptake into the HepaRG cells. Also, it appears that the hydrophilic metabolites PANOs were formed intracellularly and subsequently excreted into the cell culture medium, thus the time-dependent increase of PANOs in the supernatant.

### 2.2. Inhibition of Transporters Decelerates PA Uptake

PA and PANO content in the cell culture medium of HepaRG cells upon transport inhibition was measured by LC-MS/MS over the time period of 24 h (Figure 3). The selected PAs comprised the cyclic diesters retrorsine and senecionine as they had emerged to show adequate and measurable uptake of the mother compound and PANO formation over 24 h (Figure 2a). The applied concentration of 2.5 µM was not cytotoxic, as shown in Appendix A. 1 h, or in the case of TEA 24 h prior to PA incubation, fully differentiated cells were treated with the pharmacological transporter inhibitors that mediate a reversible chemical inhibition of transporter function.

As was observed before, retrorsine (Figure 3a,b) and senecionine (Figure 3c,d) showed a time-dependent decrease and simultaneously an increase of their respective PANOs in the cell culture medium. All transport inhibitors caused a decelerated uptake of retrorsine and senecionine over the time period of 24 h, with the exception of the very potent and selective ABCG2 inhibitor Ko143 [44], which did not alter retrorsine and senecionine uptake and PANO formation. The application of CsA, which is known to inhibit the function of a broader target spectrum including both SLC10A1 and ABCB1 [34,45,46,47], reduced retrorsine and senecionine uptake and PANO formation significantly for all time points. Also, MK-571 decelerated retrorsine and senecionine uptake and PANO excretion, but to a lesser extent than CsA. MK-571 is a selective inhibitor of some members of the ABCC family that have overlapping substrate specificities: the basolateral efflux transporter ABCC1, the apical efflux transporter ABCC2 and the basolateral exporter ABCC3 [48,49]. Qu and TEA are both used as inhibitors of the influx transporter SLC22A1 localized on the basolateral membrane of hepatocytes [50,51,52,53]. Both inhibitors attenuated retrorsine and senecionine uptake and PANO release. Qu had a striking effect on both retrorsine and senecionine uptake at all time points, whereas TEA significantly reduced retrorsine uptake only after 8 and 24 h and senecionine uptake only after 8 h.

### 2.3. Transporter Inhibition Attenuates PA-Mediated Cytotoxicity

PA-mediated cytotoxicity upon transporter inhibition was measured with the MTT assay after 24 h. Prior to PA incubation with 250 µM retrorsine and senecionine, metabolically competent HepaRG cells were pre-incubated with the pharmacological transporter inhibitors. Treatment with inhibitors alone did not reduce cell viability below 80%. A significantly lower PA-mediated cytotoxicity compared to the control group (250 µM PA only) could be detected for all inhibitors except for Ko143, which did not alter the cytotoxic effects of retrorsine and senecionine. Incubation with 250 µM retrorsine and seneconine was cytotoxic and led to a viability of 55% or 46%, respectively. In comparison, co-incubation with Qu restored viability to 100% (retrorsine) and 90% (senecionine). The other SLC22A1 inhibitor TEA rescued the viability of the PA-treated cells to 82% for retrorsine and 69% for senecionine. Similarly, CsA treatment elevated the viability to 82% for retrorsine and 63% for senecionine, respectively (Figure 4).

### 2.4. PAs Reduce SLC22A1 Substrate Uptake in a Concentration-Dependent Manner

Both Qu and TEA are rather specific inhibitors of SLC22A1 [50,51,52,53]. In combination with the specific SLC22A1 substrate and mitochondrial dye ASP^+^ that is solely transported via SLC22A1 [54], PA-mediated effects on SLC22A1 can be further assessed. HepaRG cells were incubated with retrorsine and senecionine in concentrations from 1 to 250 µM for 24 h and afterwards, ASP^+^ uptake was measured fluorometrically. The applied ASP^+^ concentration of 1 µM is not cytotoxic [51,55]. As a positive control for substrate uptake inhibition, Qu (100 µM) and TEA (3 mM) were also applied.

With increasing concentrations, both PAs reduced ASP^+^ uptake equally by up to 80%, which is more than the inhibitory effects of both Qu and TEA. In comparison to TEA, Qu is a weaker inhibitor of SLC22A1 as it reduced substrate uptake by only 18%, whereas TEA reduced substrate uptake by up to 64% (Figure 5).

## 3. Discussion

PAs have been known to pose a health risk for humans and livestock for a long time. At present, independent of their structure, all PAs are assessed as equally harmful to human health [56,57]. However, recent publications point towards the existence of different toxic potencies that seem to be in relation to their different structures [22,41,58,59,60]. As this connection is still not yet fully understood, the aim of this study was to further elucidate the toxicokinetic properties of structurally different PAs, and especially PA uptake and PANO export in the liver using a well-characterized human hepatic in vitro cell model, the human hepatoma cell line HepaRG. HepaRG cells have a metabolic capacity and transporter expression profile comparable to that of human primary hepatocytes, making them a valuable cell model for studying liver metabolism and toxicity of xenobiotics [61,62,63].

The grouping of structurally different PAs with respect to their different toxic potencies is very similar for most experimental endpoints conducted in HepaRG cells. For example, the classification of PAs according to their cytotoxic potential was confirmed with the induction of apoptosis [41], genotoxicity [58,59], as well as with the complex approach of relative potency factors that combine acute toxic, genotoxic, and cytotoxic effects in vivo and in vitro [22,64]. However, data concerning toxicokinetics have not been considered in these equations. Nevertheless, they could be useful in refining PA risk assessment as they classify the oral absorption of PAs from the intestine and their subsequent distribution, metabolism, and excretion (ADME) in the body. Transport proteins located at the basolateral and apical membrane of hepatocytes affect ADME processes and, thus, toxicokinetics in general. Unfortunately, only limited data concerning transporter-dependent uptake and disposition of PAs and PANOs are available, partly because of the many structural differences of PAs and partly because influx and efflux transporters have only recently been discovered as key contributors of PA toxicity [32,33,34]. In general, it is assumed that PAs pass the intestinal and hepatic membranes solely by facilitated biological diffusion as this is the main mechanism for many xenobiotics [65,66]. Furthermore, a study with radioactively labelled PAs showed that the majority of orally administered PAs was absorbed in the small intestine [67]. However, Hessel et al., (2014) described a distinct apical excretion of echimidine mediated by the exporter ABCB1 in the human small intestinal cell line Caco-2 and in Madin-Darby Canine Kidney cells overexpressing human ABCB1 (MDCK/hABCB1) [34]. The ABCB1-mediated diminution of bioavailable echimidine in the portal blood might explain the large discrepancies in echimidine toxicity between in vivo and in vitro studies [22,60]. Tu et al. (2013 and 2014) described the uptake of the PAs monocrotaline and retrorsine through the basolateral transporter SLC22A1 in MDCK/hSLC22A1-overexpressing cells and cultured primary rat hepatocytes. In both studies, SLC22A1 inhibitors such as TEA and Qu reduced the uptake of monocrotaline and retrorsine and attenuated their cytotoxic effects in primary rat hepatocytes. Furthermore, both PAs inhibited the uptake of a SLC22A1-specific substrate in MDCK/hSLC22A1-overexpressing cells [32,33].

As a first step to study the general mode of action of PA transport processes, PA uptake was measured under physiological (37 °C) and non-physiological (4 °C) conditions assuming that energy-driven processes are limited at 4 °C. A substantially lower uptake of compound over the course of 24 h was observed for all seven structurally different PA representatives when incubated at 4 °C, indicating a primarily active and energy-dependent uptake mechanism for all PAs. Additionally to the mother compound, the low amount of PANOs in the cell culture medium at 4 °C can be explained by both reduced uptake and metabolism resulting in extenuated PANO formation. The majority of uptake transporters in the hepatocyte membrane are SLC transporters. They are either uniporters that operate non-actively via facilitated biological diffusion (passive transport) to transport a substrate downhill its concentration gradient, or they act as co-transporters that utilize downhill ion gradients to transport both ions and substrates together either in the same (symport) or in the opposite direction (antiport) [24,27,28]. In the context of secondary active transport, the influence of temperature is indirect. In contrast to primary active transport where energy is provided directly by e.g., hydrolysis of ATP, secondary active transport is driven by an electrochemical potential difference. By letting the driving ion (e.g., Na^+^) move downhill its electrochemical gradient, energy is gained to power the transport of a second molecule. In order to maintain the gradient of the driving ion, primary active pumps such as the sodium-potassium-adenosine triphosphatase (Na^+^/K^+^-ATPase) are needed to pump ions across the cell membrane [68]. At 4 °C, the Na^+^/K^+^-ATPase would be non-functional, which, in turn, would interfere with the Na^+^ distribution and thereby limit secondary active uptake processes, too. However, low temperatures also decrease membrane fluidity and permeability which in turn also affect passive diffusion [40,69]. Therefore, the temperature method does not allow for a precise quantification of the actual share of active transport processes involved. Rather, the uptake of PAs into the cells is best explained by a combination of both passive and active uptake processes with emphasis on secondary active uptake mediated via SLC transporters. This is not uncommon, as total cellular uptake of a component is often composed partly of passive diffusion and partly of active carrier-mediated components [52,70].

In this study, a structure-dependency of the uptake of PAs into HepaRG cells was observed. This grouping coincides with other endpoints such as PA-mediated cytotoxicity [41], suggesting a selective uptake of PAs to be a prerequisite for structure-dependent toxicity. Thereby, the necine base plays a secondary role in cellular uptake and PA-mediated toxicity as both senecionine and heliosupine show high uptake and comparably strong toxic effects but belong to the retronecine- and heliotridine-type, respectively. Also, intermedine, lycopsamine, retrorsine, and senecionine all belong to the retronecine-type, but the differences in cellular uptake and cytotoxicity in HepaRG cells are evident. Rather, the necic acid is crucial in determining the uptake and toxic potential of PAs, as monoesters are not or only slightly taken up into the hepatocytes and are also considered to be less potent than diesters, which show medium (e.g., senkirkine, echimidine, retrorsine) to strong (e.g., heliosupine and senecionine) uptake and toxicity (see Figure 2a and [41]). It can be assumed that PAs with structurally similar necic acids may show a similar uptake pattern and comparable toxicity.

All inhibitors caused a reduced uptake of retrorsine and senecionine (Figure 3) and were able to attenuate the cytotoxic effects of retrorsine and senecionine (Figure 4) with the exception of ABCG2-inhibitor Ko143, excluding this apical efflux transporter to be involved in PA and PANO disposition. In contrast, the multi-target inhibitor CsA reduced both retrorsine and senecionine uptake and PANO formation, as well as their cytotoxic effects significantly. This emphasizes the inhibitor target SLC10A1 to be relevant for PA uptake. Simultaneously, the other target protein ABCB1 appears to participate in the excretion of PANOs in HepaRG cells. However, it has to be considered that CsA also inhibits CYP3A4. This enzyme is thought to be especially relevant for PA metabolism [71]. Its inhibition might be a main reason for the observed reduced PA-mediated cytotoxicity. Likewise, MK-571 application minimized retrorsine and senecionine uptake and PANO excretion as well as PA-mediated cytotoxicity, but to a lesser extent than CsA. MK-571 inhibits ABCC efflux transporters either located at the basolateral (ABCC1 and 3) or apical (ABCC2) hepatocyte membrane. Generally, ABCC transporters are efflux pumps mediating the excretion of xenobiotics and their metabolites. Therefore, it is likely that they participate in PANO elimination, too. However, no alteration in PANO content can be detected. Rather, the results are best explained by yet unknown inhibitory effects of MK-571 on other transporters and/or xenobiotic metabolizing enzymes. The SLC22A1 inhibitors Qu and TEA both reduced retrorsine and senecionine uptake and PANO excretion significantly. Also, both inhibitors rescued cytotoxic effects caused by retrorsine and senecionine. For the PAs retrorsine and monocrotaline a SLC22A1-mediated uptake has been reported before [32,33]. Further analysis of SLC22A1-mediated import of retrorsine and senecionine revealed that both PAs at a minimum concentration of 10 µM inhibited the uptake of the SLC22A1-specific substrate ASP^+^ (Figure 5). This shows that retrorsine and senecionine are both inhibitors and substrates at least for SLC22A1. Substrates that are lipophilic and have a positive net charge are more likely to be inhibitors of SLC22A1 funtion. Also, a molecular size below 500g/mol is important for SLC22A1-mediated uptake [52]. All requirements apply to both retrorsine and senecionine. However, whether these PAs inhibit the activity of SLC22A1 through direct binding or via (non-) competitive inhibition remains to be clarified.

In summary, SLC transporters such as SLC10A1 and SLC22A1 are assumed to be relevant for the uptake of retrorsine and senecionine in HepaRG cells. SLC22A1 has been reported before to be involved in PA uptake [32,33]. At least for SLC22A1, retrorsine and senecionine are both inhibitors and substrates. By contrast, the ABC efflux pump ABCG2 is excluded for the disposition of retrorsine and senecionine and their PANOs. Furthermore, Tu et al. (2013) excluded an ABCB1- and ABCG2-mediated transport of monocrotaline, as MDCK cells stably overexpressing these transporters showed no significant uptake of this PA [33].

Hepatic transporters are relevant for PA toxicokinetics as they mediate the import of PAs into the hepatocytes, which is a necessary step for PAs to exert their toxic potential. Altering influx processes, e.g., by applying transport inhibitors, can reduce the uptake of PAs and thus decrease PA-mediated cytotoxicity. As the uptake of PAs is structure-dependent, it can be gathered that a main reason for the different toxicities of structurally different PAs is their uptake. Therefore, uptake and transport mechanisms of PAs need to be further investigated.

## 4. Materials and Methods

### 4.1. Chemicals

The PAs echimidine (Em), heliosupine (Hs), intermedine (Im), lycopsamine (Ly), retrorsine (Re), senecionine (Sc) and senkirkine (Sk) as well as their respective PANOs echimidine *N*-oxide (EmNO), heliosupine *N*-oxide (HeNO), intermedine *N*-oxide (ImNO), lycopsamine *N*-oxide (ImNO), retrorsine *N*-oxide (ReNO) and senecionine *N*-oxide (ScNO, all >95% purity) were obtained from PHYTOPLAN Diehm & und Neuberger GmbH (Heidelberg, Germany) and dissolved in 50% acetonitrile (ACN)/50% water (*v*/*v*) to obtain 5 mM stock solutions. 4-(4-dimethylaminostyryl)-N-methylpyridinium iodide (ASP^+^) was purchased from Thermo Fisher Scientific Inc. (Waltham, MA, USA), 3-(4,5-dimethylthiazol-2-yl)-2,5-diphenyltetrazolium bromide (MTT) was obtained from Biomol (Hamburg, Germany) and quinidine (Qu) was purchased from Tocris Bio-Techne GmbH (Wiesbaden, Germany). All other chemicals were received from Sigma-Aldrich (Taufkirchen, Germany). All inhibitors except for tetraethylammonium bromide (TEA) were dissolved in 100% dimethyl sulfoxide (DMSO) to obtain 10 mM stock solutions for cyclosporine A (CsA) and Qu or 50 mM stock solutions for Ko143 and MK-571, respectively. TEA was dissolved in water to obtain a 100 mM stock solution.

### 4.2. Cell Culture

The human hepatoma cell line HepaRG was obtained from Biopredic International (Saint-Grégoire, France). After seeding, the cells were cultivated for two weeks in William’s Medium E with stable glutamine supplemented with 10% fetal bovine serum (FBS), 5 µg/mL human insulin (medium and both supplements from PAN-Biotech, Aidenbach, Germany), 50 µM hydrocortisone hemisuccinate (Sigma-Aldrich, Taufkirchen, Germany), 100 U/mL penicillin and 100 µg/mL streptomycin (Capricorn Scientific, Ebsdorfergrund, Germany) at 37 °C in a humidified atmosphere of 5% CO_2_. After two weeks of proliferation, differentiation was initiated by adding 1.7% of the differentiating agent DMSO [72,73,74]. After additional two weeks of cultivation, HepaRG cells were fully differentiated and could be used for four weeks in which they maintain their differentiated status [75,76]. All experiments were performed with differentiated cells seeded at passages 16 to 20 in which no marked functional variations of enzymes are known [75].

### 4.3. Inhibition Assay with Pharmacological Inhibitors

For comparison of effects of PAs on cells with non-inhibited and inhibited hepatic transporters, the following pharmacological inhibitors targeting selected efflux and influx transporters were applied to HepaRG cells 1 h prior to PA incubation: 10 µM CsA, 5 µM Ko143, 50 µM MK-571, 100 µM Qu and 3 mM TEA. In the case of TEA, cells were pre-incubated 24 h prior to PA treatment. The inhibitors were applied anew at PA incubation. All inhibitors were dissolved in 100% DMSO and dilutions were chosen so that the final DMSO concentration in the cell culture medium never exceeded 1.7%.

### 4.4. Cell Viability Assay

To analyze the cytotoxic effects of PAs on HepaRG cells with non-inhibited and inhibited hepatic transporters, the cells were seeded at a density of 9000 cells per well in the inner 60 wells of a 96-well plate and cultivated as described in Section 4.2. After four weeks of cultivation, FBS concentration in the cell culture medium was reduced to 2% for 48 h prior incubation. Thereby, inhibitory effects of peptide growth factors from the FBS on xenobiotic-metabolizing enzymes were minimized [77]. Simultaneously, the high concentration of 1.7% DMSO was upheld in order to maintain the drug-metabolizing capacity of the HepaRG cells [75]. The cells were pre-incubated with pharmacological inhibitors as described in Section 4.3. Afterwards, the cells were treated with 250 µM senecionine or retrorsine for 24 h, respectively. 2.5% ACN was used as solvent control and 0.01% Triton-X-100 served as a metabolism-independent positive control for cytotoxicity. Cell viability was analyzed via the MTT test [78]. 10 µl MTT reagent (5 mg/mL MTT dissolved in phosphate-buffered saline, PBS) was added per well and incubated for 1 h at 37 °C. The supernatant was discarded, and the violet formazan crystals dissolved in 130 µL desorption solution (0.7% sodium dodecyl sulfate in propan-2-ol). After shaking for 20 min (under protection from light), absorbance was measured on a TecanM200Pro spectrometer (Tecan Group Ltd., Männedorf, Switzerland) at a wavelength of 570 nm, along with a reference wavelength of 630 nm.

### 4.5. Transport Analysis of PAs in HepaRG Cells

In order to measure time-dependent depletion of PA and PANO content in cell culture medium, cells were seeded at a density of 0.2 × 106 cells per well in a 6-well plate and cultivated as described in Section 4.2. After four weeks of cultivation, cells were adapted to serum-free medium and medium substituted with insulin, transferrin, selenium (ITS) serum extender (Capricorn Scientific, Ebsdorfergrund, Germany) 48 h prior to PA incubation for optimal LC-MS/MS measurements. HepaRG cells were either treated with 700 nM echimidine, heliosupine, intermedine, lycopsamine, retrorsine, senecionine, or senkirkine, incubated at physiological 37 °C or at 4 °C. Samples were taken after 2 h, 4 h, 8 h and 24 h. When the inhibition assay was performed, HepaRG cells were treated with pharmacological inhibitors as mentioned in Section 4.3., thereafter incubated with 2.5 µM senecionine and retrorsine and samples were taken after 4 h, 8 h, and 24 h prior to LC-MS/MS analysis.

### 4.6. Analysis of PA and PANO Content in Cell Culture Supernatant Using LC-MS/MS

The content of echimidine, echimidine *N*-oxide heliosupine, heliosupine *N*-oxide, intermedine, intermedine *N*-oxide, lycopsamine, lycopsamine *N*-oxide, retrorsine, retrorsine *N*-oxide, senecionine, senecionine *N*-oxide, and senkirkine was analyzed by LS-MS/MS following the methodology mentioned in Kaltner et al. (2019). Briefly, chromatographic separation of PA/PANO analytes was achieved on a 50 × 2.1 mm Kinetex ™ 2.6 μm CoreShell EVO C18 100 Å column protected by a SecurityGuard ™ ULTRA EVO C 18 2.1 mm guard column (both Phenomenex, Aschaffenburg, Germany) at a flow rate of 0.4 mL/min on a Shimadzu high performance liquid chromatography (HPLC) apparatus (LC-20AB, SIL-20AC HT, CTO-20AC, CBM-20A, Duisburg, Germany) coupled to an API4000 triple quadrupole MS (Sciex, Darmstadt, Germany). The following MS parameters were used for measurements in positive electrospray ionisation (ESI+) mode: ionisation voltage 2500 V, nebuliser gas 50 psi, heating gas 50 psi, curtain gas 30 psi, temperature 600 °C, collision gas level 7. The substance specific parameters are summarized in Kaltner et al. 2019 [79]. A calibration curve covering a range from 10 to 125 nM of PAs and respective PANOs was used for quantification. Depending on the experiment, measured PA and PANO contents in the cell culture medium were referred to the initial PA concentration of 700 nM (in case of echimidine, heliosupine, intermedine, lycopsamine, retrorsine, senecionine, senkirkine) or 2.5 µM (in case of retrorsine and senecionine) at time point zero (t_0_). Representative chromatograms of retrorsine, retrorsine *N*-oxide, senecionine and senecionine *N*-oxide are shown in Appendix A.

### 4.7. ASP^+^ Uptake Measurements

ASP^+^ is an organic cationic mitochondrial dye that only fluoresces inside cells so that fluorescence measured in the cell culture medium can be neglected [51]. For measuring the uptake of ASP^+^ in HepaRG cells incubated with PAs, cells were treated with retrorsine and senecionine in concentrations ranging from 1-250 µM. In parallel, cells were incubated with Qu (100 µM) and TEA (3 mM) as mentioned in Section 4.3. After 24 h, cells were incubated with 1 µM ASP^+^ in Hanks’ balanced salt solution (HBSS, Sigma-Aldrich, Taufkirchen, Germany) at 37 °C. Fluorescence was measured after 20 min at ASP^+^-specific wavelength (λex = 475 nm and λem = 605 nm) on a TecanM200Pro spectrometer (Tecan Group Ltd., Männedorf, Switzerland) according to Ahlin et al., (2008) [51]. 

### 4.8. Statistical Analysis

Statistical analysis was performed using Sigmaplot 14.0 software (Systat Software, Erkrath, Germany). Statistically significant differences in multi-comparison procedures with one variable (e.g., different inhibitors during one time point) were calculated by one-way analysis of variance (ANOVA) followed by Dunnett’s post-hoc test versus the corresponding solvent control. Results were considered as significant at *p* < 0.05 and are indicated in the graphs by * *p* < 0.05, ** *p* < 0.01, *** *p* < 0.001.

## Figures and Tables

**Figure 1 ijms-22-03821-f001:**
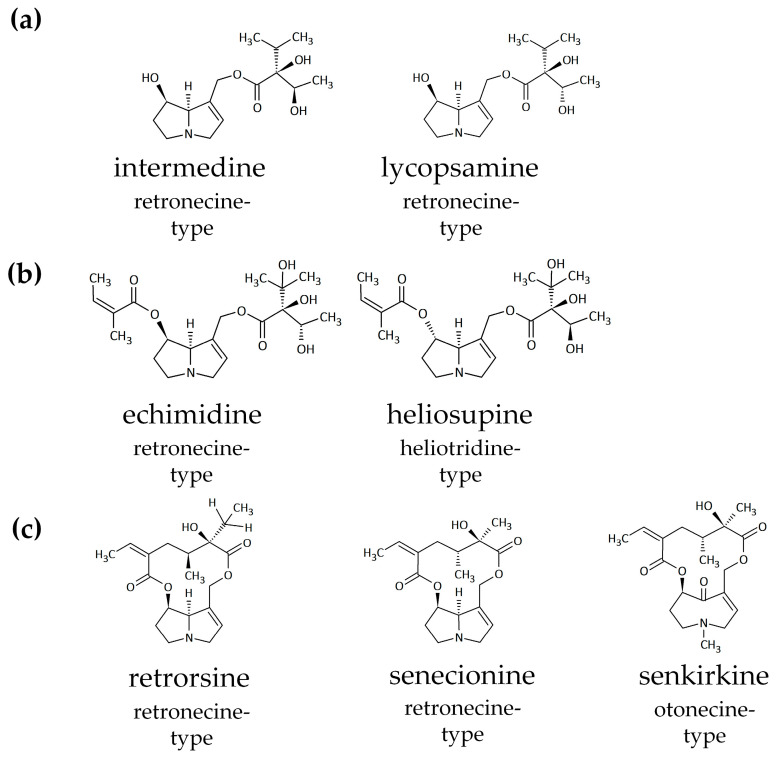
Chemical structures of 1,2-unsaturated pyrrolizidine alkaloids (PAs) used in this study, representing the main structural characteristics of their necine base (retronecine-type, heliotridine-type and otonecine-type) and of their necic acids: (**a**) monoester, (**b**) open-chain diester, and (**c**) cyclic diester.

**Figure 2 ijms-22-03821-f002:**
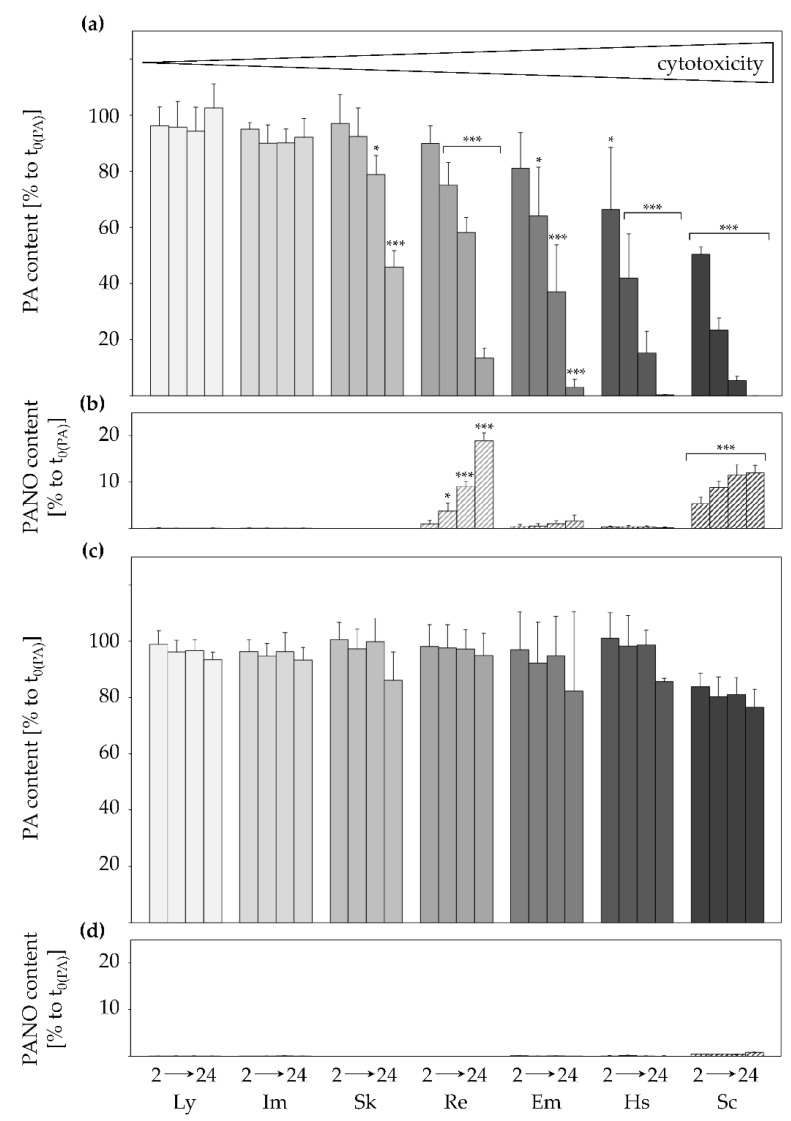
Time-dependent reduction of seven structurally different PAs in cell culture medium of differentiated HepaRG cells under different temperatures of 37 °C (**a**,**b**) and 4 °C (**c**,**d**). HepaRG cells were incubated for 2, 4, 8, and 24 h with 700 nM of lycopsamine, intermedine, senkirkine, retrorsine, echimidine, heliosupine, or senecionine, respectively. Cell culture medium was collected after each time point and PA content (gray bars) and PANO content (striped bars) measured by LC-MS/MS. PA and PANO content is given in percent of the initial concentration of respective PA (at t_0_) as mean + SD of three to four independent experiments with two incubations each. Please note the unequal scaling of the different y-axes. Cytotoxicity grading was done using data from Glück et al. (2021) [41]. For raw data presentation in ng/mL please see Appendix A. Statistical analysis was performed using one-way ANOVA, followed by Dunnett’s post-hoc test, and significant differences are depicted as follows: * for PA and PANO content with * *p* < 0.05, *** *p* < 0.001. Ly, lycopsamine; Im, intermedine; Sk, senkirkine; Re, retrorsine; Em, echimidine; Hs, heliosupine; Sc, senecionine.

**Figure 3 ijms-22-03821-f003:**
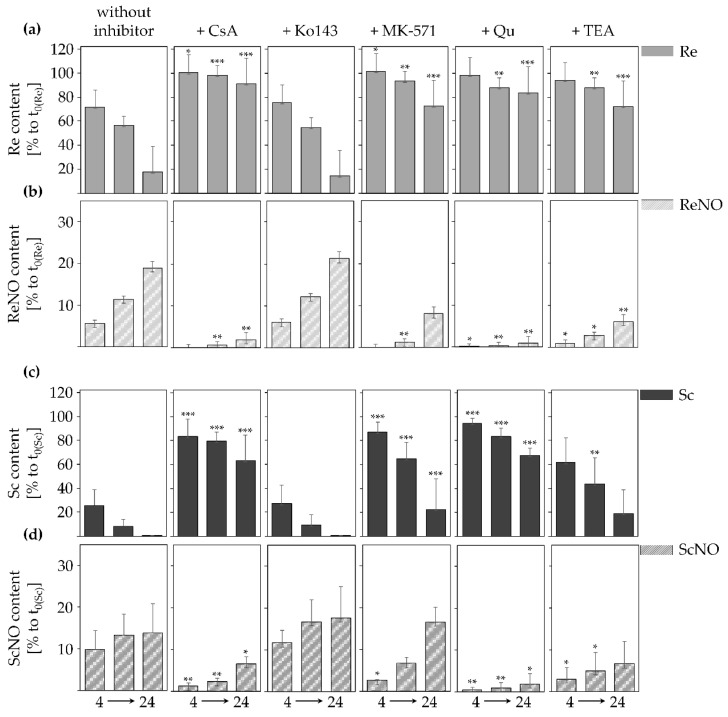
Effects of pharmacological influx and efflux inhibitors on time-dependent reduction of retrorsine (**a**,**b**) and senecionine (**c**,**d**) in cell culture medium of differentiated HepaRG cells. HepaRG cells were pre-incubated for 1 h with the inhibitors CsA (10 µM), Ko143 (5 µM), MK-571 (50 µM), Qu (100 µM) and for 24 h with TEA (3 mM) before being incubated with 2.5 µM retrorsine or senecionine, respectively. Cell culture medium was collected after 4, 8, and 24 h and PA content (gray bars) and PANO content (striped bars) measured by LC-MS/MS. PA and PANO content is given in percent of the initial concentration of respective PA (at t_0_) as mean + SD of three to five independent experiments with two incubations each. Please note the unequal scaling of the different y-axes. For raw data presentation in ng/mL please see Appendix A. Statistical analysis was performed using one-way ANOVA, followed by Dunnett’s post-hoc test, and significant differences for each inhibitor in relation to the respective time point of the control group are depicted as follows: * for PA and PANO content at one time point for inhibitor-treatment vs. no-inhibitor with * *p* < 0.05, ** *p* < 0.01, *** *p* < 0.001. Re, retrorsine; ReNO, retrorsine *N*-oxide; Sc, senecionine; ScNO, senecionine *N*-oxide.

**Figure 4 ijms-22-03821-f004:**
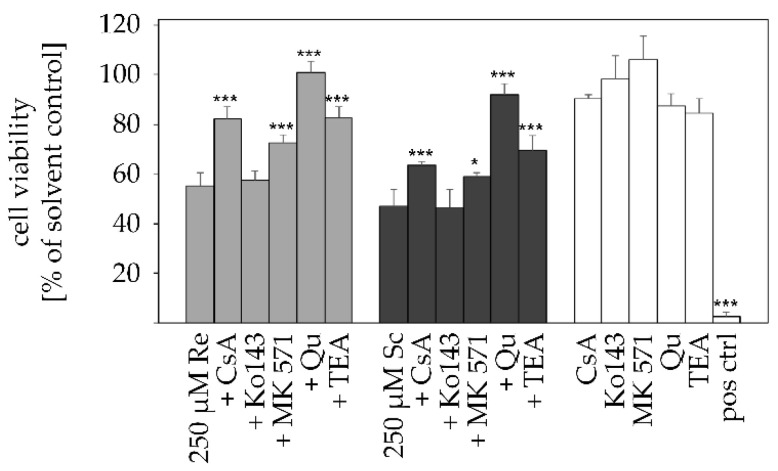
Effects of pharmacological influx and efflux inhibitors on retrorsine- and senecionine-induced cytotoxicity in differentiated HepaRG cells. HepaRG cells were pre-incubated for 1 h with the inhibitors CsA (10 µM), Ko143 (5 µM), MK-571 (50 µM), Qu (100 µM) or for 24 h with TEA (3 mM) before being incubated with 250 µM retrorsine (bright gray bars) or senecionine (dark gray bars), respectively. Simultaneously, all inhibitors were incubated for 24 h as positive controls, respectively (white bars). Cell viability was assessed using the MTT assay and is displayed as percent of solvent control as mean + SD of three to thirteen independent experiments with five incubations each. 0.01% Triton X-100 was used as a positive control (pos ctrl), resulting in a decrease of cell viability below 5%. Statistical analysis was performed using one-way ANOVA, followed by Dunnett’s post-hoc test, and significant differences are depicted as follows: * *p* < 0.05, *** *p* < 0.001. Re, retrorsine; Sc, senecionine.

**Figure 5 ijms-22-03821-f005:**
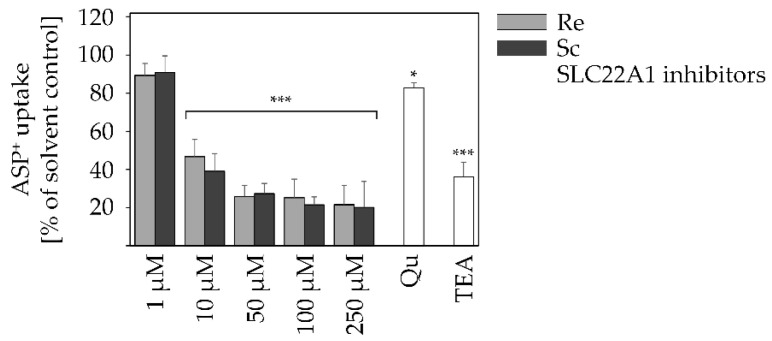
Reduction of fluorescent SLC22A1 substrate (ASP^+^) uptake in retrorsine- and senecionine-incubated HepaRG cells in a concentration-dependent manner. Differentiated HepaRG cells were incubated for 24 h with different concentrations (1–250 µM) of retrorsine (bright gray bars) or senecionine (dark gray bars), respectively. The SLC22A1 inhibitors Qu and TEA were incubated for 24 h as positive controls at 100 µM or 3 mM, respectively (white bars). ASP^+^-uptake is displayed as percent of solvent control as mean + SD of six independent experiments with five incubations each. 0.01% Triton X-100 was used as positive control for cytotoxicity, resulting in a decrease of cell viability below 5%. Statistical analysis was performed using one-way ANOVA, followed by Dunnett’s post-hoc test, and significant differences are depicted as follows: * *p* < 0.05, *** *p* < 0.001. Re, retrorsine; Sc, senecionine.

## Data Availability

The data presented in this study are available in the Appendix A which can be found at https://www.mdpi.com/ethics.

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
