# Peer review of "Active Transport of Hepatotoxic Pyrrolizidine Alkaloids in HepaRG Cells"

_ijms, 2021, doi:10.3390/ijms22083821_

Round 1
Reviewer 1 Report
The manuscript entitled “Uptake and transport of hepatotoxic pyrrolizidine alkaloids in HepaRG cells” describes the participation of drug-transporters on the import of two alkaloids into cells and the export of their N-oxides from the HepaRG cells. The topic and some results of the MS are interesting, but I have certain reservations about the presentation of the data. In addition, I miss the results of some important experiments. I suggest following changes and amendments:
- Since all experiments (except the first one - the comparison of the transport of the alkaloids at 37°C and 4°C) were done only with retrorsine ( Re) and senecionine (Sc) I recommend focusing only on these two alkaloids and withdrawing the others from Fig.1 and Fig.2.
- Change the title to “Active transport of retrorsine and senecionine in HepaRG cells”
- Add the results of identification and quantification of all metabolites of Re and SC in HepaRG to illustrate the prevalence of N-oxides (for explanation why only N-oxides were analyzed)
- Add the results of the test of toxicity of Re and Sc in HepaRG cells to demonstrate that nontoxic concentrations were used in transport experiments
5.Fig.2 and Fig 3 - present the amounts of PA and PA-NO separately to be clearer
6.Fig 3 - use designation “without inhibitors” instead of 2.5uM PA
- Fig.4 - present also the viability of the cells exposed to individual inhibitors only (without PA) as these compounds in such high concentrations might affect it
- Fig.5 - present only the results of ASP uptake in this figure to be clearer
I hope these changes improve the MS to be acceptable in IJMS
Reviewer 2 Report
The authors presented a research article on the “Uptake and transport of hepatotoxic pyrrolizidine alkaloids in 2 HepaRG cells”.
The topic is interesting and well within the aims and scopes of the Journal.
Yet, the manuscript needs some implementations and changes before it can be accepted for publication in this Journal.
The things to change and add are listed below one by one:
INTRODUCTION:
- Line 54: In the compound name H must be written in Italics.
RESULTS:
- Please explain all your results from the chemical and biological standpoints.
MATERIALS AND METHODS:
- Could you not study the same effects starting from compounds extracted from the plants? Why not? I think that would also be interesting since you know the possible different effects obtained with natural and synthetic compounds.
- More information must be provided about the LC-MS/MS instrument and instrumental settings
REFERENCES:
- All the references are not fully written as requested by the Journal.
Round 2
Reviewer 1 Report
- I have not changed my opinion and I recommend focusing on this MS only on Re and Sc.
- I understand that quantification of all Re and Sc metabolites is not possible, but I insist on presenting typical chromatograms with metabolite peaks (should be labeled N-oxides) and the parent compounds.
- In Figures 2 and 3, the data should be given in units of concentration, not in percentages.
- Figures 2 and 3 should show four standard graphs (a, b, c, d - two for Re and Sc, two for their N-oxides), each with a description of the y-axis
- Figure 4. - Combine these three graphs into one.
Reviewer 2 Report
The authors presented a revised version of the manuscript I have previously reviewed.
The authors satisfactorily addressed all my major queries. Thus, the manuscript can be accepted in its present form even if the style of the reference is still not completely the one requested by the Journal, but this thing can be adjusted during the proof-reading.
Author Response
We thank reviewer 2 for his comment. We revised the reference list especially in regard to the abbreviations of the jourmals of the cited references.
Round 3
Reviewer 1 Report
All data in figures must be in molar concentrations!